# UNSUPERVISED ADVERSARIAL IMAGE RECONSTRUCTION

**Arthur Pajot**[*][a]**, Emmanuel de Bézenac**[*][a]**, Patrick Gallinari**[a, b]
{arthur.pajot, emmanuel.de-bezenac, patrick.gallinari}@lip6.fr
[a] Sorbonne Universités, UMR 7606, LIP6, F-75005 Paris, France
[b] Criteo AI Lab, Paris, France

## ABSTRACT

We address the problem of recovering an underlying signal from lossy, inaccurate observations in an unsupervised setting. Typically, we consider situations where there is little to no background knowledge on the structure of the underlying signal, no access to signal-measurement pairs, nor even unpaired signal-measurement data. The only available information is provided by the observations and the measurement process statistics. We cast the problem as finding the *maximum a posteriori* estimate of the signal given each measurement, and propose a general framework for the reconstruction problem. We use a formulation of generative adversarial networks, where the generator takes as input a corrupted observation in order to produce realistic reconstructions, and add a penalty term tying the reconstruction to the associated observation. We evaluate our reconstructions on several image datasets with different types of corruptions. The proposed approach yields better results than alternative baselines, and comparable performance with model variants trained with additional supervision.

## 1 INTRODUCTION

Many real world applications require acquiring information about the state of some physical system from incomplete and inaccurate measurements. For example, in infrared satellite imagery, one has to deal with the presence of clouds and a variety of other external factors perturbing the acquisition of temperature maps. This raises questions on how to recover the correct information and eliminate the contribution of external factors hindering the overall signal acquisition.

In this context, signal recovery does not usually yield a unique solution, meaning that multiple signal reconstructions could trivially explain the measurements. For the above example, different missing temperature values could accurately explain the observations. To cope with this indeterminacy, one usually relies on prior information on the structure of the true signal in order to constrain the reconstruction to plausible solutions (Stuart (2010)). A common approach is to use handcrafted, analytically tractable priors (Candès et al. (2005), Mota et al. (2017)). This approach is limited to situations for which the underlying signal structure can be easily described, which are rarely observed in the wild.

Recent developments in generative models parameterized by neural networks (Goodfellow et al. (2014), Kingma & Welling (2013), Dinh et al. (2016)) offer a promising statistical approach to signal recovery, for which priors on the signal are not handcrafted anymore, but learned from large amounts of data. Despite exhibiting interesting results (Bora et al. (2017), Mardani et al. (2017), Ledig et al. (2016)), these methods all require some form of supervision, either observation measurement-signal pairs, or at least unpaired samples from observations and underlying signals. For many practical problems, obtaining these samples is too expensive and/or impractical, which makes these approaches not suitable for such situations.

We address the problem of image reconstruction in an unsupervised setting, when only corrupted observations are available, together with some prior information on the nature of the measurement process.

---

[*]equal contribution

The learning problem is formulated as finding the maximum a posteriori estimate of signals given their measurements on the training set. We derive a natural objective for our reconstruction network, composed of a linear combination of an adversarial loss for recovering realistic signals, and a reconstruction loss to tie the reconstruction to its associated observation (Section 2.2). This model is evaluated and compared to baselines on 3 image datasets, CelebA (Liu et al. (2015)), LSUN Bedrooms (Yu et al. (2015)), Recipe-1M (Marin et al. (2018)), where we experiment with different types of measurement processes corrupting the images.

Our contributions are:

- A novel, computationally efficient framework for dealing with large scale signal recovery in an unsupervised context, applicable to a wide range of situations,
- A model and a new way of training a deep learning architecture for implementing this framework,
- Extensive evaluations on a number of image datasets with different measurement processes.

## 2 Preliminaries

**Notations.** We use capital letters (*e.g.* $X$) for random variables, and lower-case letters (*e.g.* $x$) for their values. $\mathrm{p}_X(x)$ denotes the distribution (or its density in the appropriate context) of $X$ evaluated at $x$.

### 2.1 Problem setting.

Suppose there exists a signal $X \sim \mathrm{p}_X$ we wish to acquire, but we only have access to this signal through lossy, inaccurate measurements $Y \sim \mathrm{p}_Y$. The measurement process is modeled through a stochastic operator $F$ mapping signals $X$ to their associated observations $Y$. We will refer to $F$ as the *measurement process*, which *corrupts* the input signal. $F$ is parameterized by a random variable $\Theta \sim \mathrm{p}_\Theta$ following an underlying distribution $\mathrm{p}_\Theta$ we can sample from, which represents the factors of corruption. Thus, given a specific signal $x$, we can simulate its measurement by first sampling $\theta$ from $\mathrm{p}_\Theta$, and then computing $F(x; \theta)$. Additional sources of uncertainty, *e.g.* due to unknown factors, can be modeled using additive *i.i.d.* Gaussian noise $\mathcal{E} \sim \mathcal{N}(0, \sigma^2 I)$, so that the overall observation model becomes:

$$Y = F(X; \Theta) + \mathcal{E} \tag{1}$$

$F$ is assumed to be differentiable *w.r.t.* its first argument $X$, and $\Theta$ and $X$ to be independent (denoted $X \perp\!\!\!\perp \Theta$). Different instances of $F$ will be considered (refer to Section 4.2), like random occlusions, information acquisition from a sparse subset of the signal, overly smoothing out and corrupting the original distribution with additive noise, etc. In such cases, the factors of corruption $\Theta$ might respectively represent the position of the occlusion, the coordinates of the acquired information, or simply the values of the additive noise.

### 2.2 Approach

Given an observation $y$, our objective is to find a signal $\hat{x}$ as close as possible to the associated true signal $x$. From a probabilistic viewpoint, it is natural to formulate the problem as finding the *maximum a posteriori* (MAP) estimate, which consists in selecting the most probable signal $x^*$ under the posterior distribution $\mathrm{p}_{X|Y}(\cdot|y)$:

$$x^* = \arg\max_x \log \mathrm{p}_{X|Y}(x|y) \tag{2}$$

or equivalently:

$$x^* = \arg\max_x \log \mathrm{p}_{Y|X}(y|x) + \log \mathrm{p}_X(x) \tag{3}$$

where $p_{Y|X}(y|x)$ is the likelihood of the signal $x$ given observation $y$, and $p_X(x)$ is the prior probability evaluated at $x$. Therefore, a good reconstruction must be likely to have generated the data, *i.e.* yield high likelihood, and look realistic, *i.e.* yield high probability under the prior.

In the general case, calculating the likelihood term $p_{Y|X}(y|x)$ requires marginalizing over noise parameters $\Theta$ and this does not yield an analytic form. As for the prior $p_X(x)$, it is unknown, and we have no access to samples from $X$ since we are in an unsupervised setting: there is then no direct way to estimate $p_X$ either. In the general case considered here, with no assumption on the form of the distributions, solving Equation (3) is up to our knowledge an open problem.

In the following sections, we will introduce an approach to deal with the likelihood term (Section 3.1), and the unknown prior term (Section 3.2) in order to provide an approximate solution to equation (3) (Section 3.3). For that, we will formulate the problem as learning a mapping $G : \mathcal{Y} \rightarrow \mathcal{X}$ that links each measurement $y$ to its associated MAP estimate $x^*$ on the training set. The associated objective is then:

$$G^* = \arg\max_{G} \mathbb{E}_{p_Y} \left\{ \log p_{Y|X}(y|G(y)) + \log p_X(G(y)) \right\} \qquad (4)$$

Which is obtained by plugging $G(y) = x$ into equation 3 and taking the expectation w.r.t. the distribution of observations $p_Y$.

## 3 METHOD

From Equation (4), we see that a valid reconstruction mapping $G$ must yield high probability for the likelihood and the prior. This will guide the design of an appropriate objective during the following section, where the reconstruction mapping $G$ will be implemented using a neural network.

### 3.1 HANDLING THE LIKELIHOOD TERM

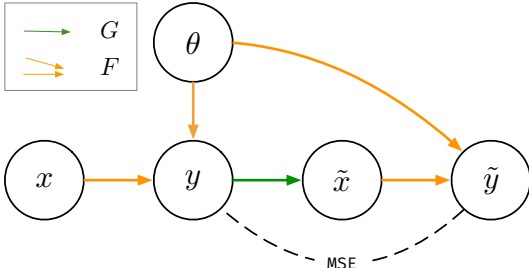

Figure 1 – The Figure illustrates the dependencies between the variables considered for handling the likelihood term when solving (4). The likelihood term in (4) can be replaced by the expectation of $\frac{1}{2\sigma^2} \|y - F(G(y); \theta)\|_2^2$ (see Equation 7). To compute this expectation, one first simulates an observation $y$ from a signal $x$ using $F(x; \theta)$, then generates $\tilde{x} = G(y)$ and $\tilde{y} = F(\tilde{x}; \theta)$, as in the Figure. This allows us to compute the MSE term in the above expression.

In the general case, evaluating the likelihood $p_{Y|X}(y|x)$ in equation (3) requires marginalizing on the unobserved noise variable $\Theta$: $p_{Y|X}(y|x) = \mathbb{E}_{p_\Theta} p_{Y|X,\Theta}(y|x,\theta)$, which involves computing an intractable integral. Most probabilistic model for image denoising make assumptions on the structure of the measurement operator $F(., \Theta)$ and on the distribution of $\Theta$ in order to obtain an analytic form for the expectation (Boyat & Joshi (2015), Alkinani & El-Sakka (2017)). Here, we consider more general measurement operators which do not necessarily lead to such a simplification and therefore proceed in a different way.

We outline below, the main steps of the method for handling the likelihood term $p_{Y|X}(y|x)$ in equation (4). The complete derivation is provided in Appendix A.

1. Making use of the independence between $X$ and $\Theta$, the expectation term $\mathbb{E}_{\mathrm{p}_Y} \log \mathrm{p}_{Y|X}(y|G(y))$ in equation (4) can be rewritten as :

$$\mathbb{E}_{\mathrm{p}_\Theta \mathrm{p}_X \mathrm{p}_{Y|X,\Theta}} \log \mathrm{p}_{Y|X,\Theta}(y|G(y),\theta) + c_1 \qquad (5)$$

, with $c_1$ constant w.r.t. $G$.

2. The general measurement process described in equation (1) induces $\log \mathrm{p}_{Y|X,\Theta}(y|G(y),\theta)$ to yield a simple analytic expression:

$$\log p(y|G(y),\theta) = -\frac{1}{2\sigma^2} \|y - F(G(y);\theta)\|_2^2 + c_2 \qquad (6)$$

with $c_2$ a constant.

3. The likelihood term $\mathbb{E}_{\mathrm{p}_Y} \log \mathrm{p}_{Y|X}(y|G(y))$ can then be replaced in objective (4) by

$$-\mathbb{E}_{\mathrm{p}_\Theta \mathrm{p}_X \mathrm{p}_{Y|X,\Theta}} \frac{1}{2\sigma^2} \|y - F(G(y);\theta)\|_2^2 \qquad (7)$$

Equation (7) shows that the likelihood term can be evaluated by first sampling a measurement $y$ conditioned on a corruption parameter $\theta$ and signal $x$, and then constrain $G$ such that $\|y - F(G(y);\theta)\|_2^2$ is close to zero. Note that in this expression, the same parameter $\theta$ is used for simulating $\tilde{y}$ from $\tilde{x}$ and $y$ from $x$ (see Figure 1 and section 3.3 for more details). Unfortunately, this requires first sampling $x$ from the signal distribution $\mathrm{p}_X$ which is unknown. In the following sections, we will see how we work around this problem.

## 3.2 HANDLING THE PRIOR TERM

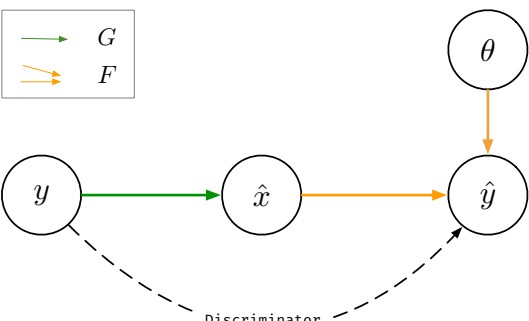

Figure 2 – The figure illustrates the dependencies of the variables used for dealing with the prior term in (4). An observation $y$ is sampled, and then transformed by the generative network into a reconstructed signal $\hat{x} = G(y)$. One then simulates a measurement $\hat{y} := F(\hat{x};\theta)$ from this reconstruction. We then enforce the distributions of observations $\mathrm{p}_Y$ and simulated measurements $\mathrm{p}_Y^G$ to be similar using an adversarial loss. In order to produce indistinguishable distributions, the generator $G$ has to *remove* the corruption and recover a sample $\hat{x}$ from $\mathrm{p}_X$.

Maximizing w.r.t. the prior term $\mathrm{p}_X(G(y))$ in equation (4) is similar to learning a mapping $G$, such that the distribution induced by $G(y)$, $\mathbb{E}_{\mathrm{p}_Y} \mathrm{p}_X(G(y))$ is close to the distribution $\mathrm{p}_X$. The prior $\mathrm{p}_X$ being unknown, the only sources of information are the lossy measurements $y$ and the known prior $p_\Theta$ on the measurement process. In order to learn an approximation of the true prior $\mathrm{p}_X$, we will use a form of generative adversarial learning, and build on an idea introduced in the AmbientGAN model by Bora et al. (2018).

AmbientGAN aims at learning an *unconditional* generative model $G$ of the true signal distribution $\mathrm{p}_X$, when only lossy measurements $y$ of the signal are available together with a known stochastic measurement operator $F$. In AmbientGAN, a generator is trained to produce uncorrupted signal samples from a latent code so that the generated signals when corrupted are indistinguishable from

the observation measurements. In Bora et al. (2018), the authors show that for some families of noise distributions $p_\Theta$, the generator's induced distribution matches the signal's true distribution. Note that even if the generation process of the observations $y$ in AmbientGAN is similar to the one considered in this paper (see Section 2.1), the objective is however different: when the aim of AmbientGAN is to learn a distribution of the underlying signal by sampling a latent space, ours is to reconstruct corrupted signals.

In order for $G$ to produce uncorrupted signals we will use an approach inspired from AmbientGAN, as illustrated in Figure 2. Given an observation $y$, one wants to reconstruct a latent signal approximation $\hat{x} = G(y)$ so that a corrupted version of this signal $\hat{y} = F(\hat{x})$ will have a distribution indistinguishable from the one of the observations $y$. The generator $G$ and a discriminator $D$ are trained on observations $y$ and generated samples $\hat{y}$. The corresponding loss is the following[1]:

$$\mathcal{L}^{\text{prior}}(G) := \max_D \ \mathbb{E}_{Y \sim p_Y, \hat{Y} \sim p_Y^G} \big\{ \log D(y) + \log \big( 1 - D(\hat{y}) \big) \big\} \tag{8}$$

where $p_Y^G$ corresponds to the distribution induced by $G$'s corrupted outputs ($\hat{y}$ in Figure 2) , *i.e.* $p_Y^G(y) := \mathbb{E}_{p_\Theta p_X^G} \big\{ p(y|x, \theta) \big\}$ and $p_X^G$ denotes the marginal distribution induced by $G$'s outputs ($\hat{x}$ in Figure 2): $p_X^G(x) := \mathbb{E}_{p_Y} p_{X|Y}^G(x|y) = \mathbb{E}_{p_Y} \delta \big( x - G(y) \big)^2$. This penalty enforces the marginal $p_X^G$ to be close to the true prior distribution $p_X$, and thus forces $G$ to map its input measurements onto $p_X$.

## 3.3 PUTTING EVERYTHING TOGETHER

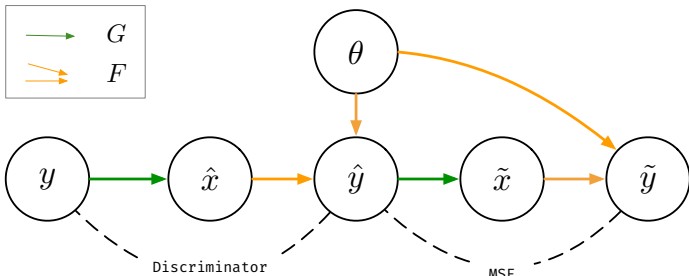

Figure 3 – General Approach. We wish to train $G$ to recover a plausible signal from lossy measurements. As is shown in Section 2.2, this requires the reconstructions $\hat{x} := G(y)$ to have high probability under the likelihood and the prior. For simplicity, variable $\mathcal{E}$ has been omitted. *Prior :* we sample a measurement $y$ from the data, produce a reconstruction $\hat{x}$, and sample a perturbation parameter $\theta$. We enforce the simulated measurement $\hat{y} := F(\hat{x}; \theta)$ to be similar to measurements in the data using an adversarial penalty. Intuitively, this requires the network to *remove* the corruption. *Likelihood :* to enforce $G$ to produce reconstructions with high likelihood, it is not possible to add a penalty to constrain the mean square error (MSE) between $y$ and $\hat{y}$ to be small. This is because the underlying perturbation that caused $y$ is unknown, and may be different from $\theta$. Starting from $\hat{y}$ we generate a $\tilde{y}$ (see figure 3) using the same $\theta$ as the one used for generating $\hat{y}$. We then constrain $\|\hat{y} - \tilde{y}\|_2^2$ to be small ($\tilde{y} = F(G(\hat{y}))$.

In Section 3.1, we have shown that it is possible to maximize the average log-likelihood term in equation (4), given that we can sample from the unknown prior distribution $p_X$. In Section 3.2, we have shown how it is possible to enforce the generator to produce samples from $p_X$, without ever having access to uncorrupted samples. The idea is then to use the distribution induced by the generator's output $p_X^G$ as a proxy for $p_X$ to compute an approximate value of the expectation in equation (5): This gives us the following penalty term (see appendix A):

$$\mathcal{L}^{\text{likeli}}(G) := \mathbb{E}_{p_\Theta p_X^G, \hat{Y} \sim p_{Y|X,\Theta}} \|\hat{y} - F(G(\hat{y}); \theta)\|_2^2 \tag{9}$$

---

[1] the min term of the adversarial loss will be introduced later, see Equation (10)

[2] $\delta(x)$ is the Dirac delta function, which is equal to zero everywhere except in $x$.

The full objective is a linear combination of penalties (8) and (9):

$$\arg\min_{G} \; \mathcal{L}^{\text{prior}}(G) + \lambda \cdot \mathcal{L}^{\text{likeli}}(G) \tag{10}$$

As illustrated by the dependencies highlighted in Figure 2, in the process of minimizing $\mathcal{L}^{\text{prior}}$, we sample from the marginal likelihood $\mathrm{p}_Y^G(y) := \mathbb{E}_{\mathrm{p}_\Theta \mathrm{p}_X^G}\{\mathrm{p}(y|x,\theta)\}$. The expectancy in the likelihood term $\mathcal{L}^{\text{likeli}}$ is precisely computed w.r.t. this distribution. We can then use the same samples in order to minimize the full objective (10). This gives us the Algorithm 1 described below, along with the dependency structure illustrated in Figure 3.

---

**Algorithm 1** Training Procedure.

---

**Require:** Initialize parameters of the generator $G$ and the discriminator $D$.

  **while** $(G, D)$ not converged **do**

      Sample $\{y_i\}_{1 \leq i \leq n}$ from data distribution $\mathrm{p}_Y$

      Sample $\{\theta_i\}_{1 \leq i \leq n}$ from $P_\Theta$

      Sample $\{\varepsilon_i\}_{1 \leq i \leq n}$ from $P_\mathcal{E}$

      Set $\hat{y}_i$ to $F(G(y_i), \theta_i) + \varepsilon_i$ for $1 \leq i \leq n$

      Update $D$ by ascending:

$$\frac{1}{n}\sum_{i=1}^{n} \log D(y_i) + \log(1 - D(\hat{y}_i))$$

      Update $G$ by descending:

$$\frac{1}{n}\sum_{i=1}^{n} \lambda \cdot \|\hat{y}_i - F(G(\hat{y}_i); \theta_i)\|_2^2 + \log(1 - D(\hat{y}_i))^3$$

  **end while**

---

## 4 EXPERIMENTS

### 4.1 MODEL ARCHITECTURES AND DATASETS

**Architectures.** We will briefly describe the architectures, additional details on architectures and hyperparameters can be found in appendix B. Our network architectures are inspired by the GAN architecture in Zhang et al. (2018). We use the same discriminator, and we propose an image-to-image variant of their latent-to-image generator for the reconstruction network $G$.

**Datasets.** We evaluate our approach using three different image datasets :

- **CelebA.** Dataset of celebrities, containing approximately 200 000 samples. As Bora et al. (2018), the images are center-cropped.

- **LSUN Bedrooms.** Dataset of bedrooms, containing 3 million samples.

- **Recipe-1M.** Dataset of cooked meals, containing approximately 600 000 samples.

All the images have been resized to $64 \times 64$. In order to place ourselves in the most realistic setting possible, every image has been corrupted once, *i.e.* there is never multiple occurrences of an image corrupted with different corruption parameters.

We withhold 15% of the training set for validation, selected uniformly at random for each dataset.

---

[3]In practice we optimize $-\log D(\hat{y}_i)$ instead of $\log(1 - D(\hat{y}_i))$

## 4.2 CORRUPTIONS

Let us present the different measurement processes $F$ used in the experiments, also named corruptions:

**Remove-Pixel.** This measurement process randomly samples a fraction $p$ of pixels uniformly and sets the associated channel values to $0$. All the corresponding channel values are set to $0$.

**Remove-Pixel-Channel.** Instead of setting to $0$ a pixel for all channels as in Remove-Pixel, one samples a pixel coordinate and a channel, and sets the corresponding value to $0$.

**Convolve-Noise.** Here $F(x; \theta) := k * x + \theta$, where $*$ is the convolution operator and $k$ is a mean filter of size $l$. For each pixel, noise $\theta$ sampled from a zero-mean Gaussian of variance $\sigma_C^2$ is added to the previous result.

**Patch-Band.** A horizontal band of height $h$ whose vertical position in the image is uniformly sampled from the set of possible positions. For each pixel falling inside the band, its associated value is set to $0$. The resulting measurement for pixel at column $i$ and row $j$ can be summarized as:

$$F(x; \theta)_{i,j} := \begin{cases} 0, & \text{if } j \in \{\theta, \ldots, \theta + h\} \\ x_{i,j}, & \text{otherwise} \end{cases} \tag{11}$$

where $\theta$ is uniformly sampled from $\{1, \ldots, H - h\}$, and $H$ is the image height. In the experiments, $h$ is set to $20$.

## 4.3 BASELINES

### 4.3.1 CONDITIONAL AMBIENTGAN.

This is our only unsupervised baseline. The context is the same as for our model: the measurement process $F$ is assumed known, there is no access to samples from the uncorrupted signal distribution $p_X$, but only to their corrupted counterpart $p_Y$. This baseline is a combination of two recent techniques in the field of signal recovery: the aforementioned AmbientGan Bora et al. (2018) and CS-GAN Bora et al. (2017).

An unconditional generator $G$ is trained using the AmbientGan framework (Bora et al. (2018)) for each type of measurement process $F$, in order to produce samples from $p_X$ (see Section 4.2). The distribution induced by the generator $p_X^G$ is an approximation of $p_X$ (at the optimum, both distributions match, *i.e.* $p_X^G = p_X$). Given a specific measurement $y$, the reconstruction $\hat{x}$ is the signal from $G$ that is closest to $y$, as in Bora et al. (2017). To find $\hat{x} = G(\hat{z})$, we search for the latent code $\hat{z}$ of $G$, such that $\hat{z} = \arg\min_z \|y - G(z)\|_2^2 + R(z)$. $R(z)$ is a regularizing term that enforces the latent code to stay in $G$'s input domain. This objective is optimized using stochastic gradient descent. To train $G$, we use the same architectures and hyper-parameters as those provided by the authors. Because this approach may be sensitive to the initial latent code, we reiterate this approach three times and select the best resulting image.

### 4.3.2 UNPAIRED VARIANT.

This is a variant of our model where we have access to samples of the signal distribution $p_X$. This means that although we have no paired samples from $p_{X,Y}$, we have access to unpaired samples from $p_X$ and $p_Y$. This baseline is similar to our model but instead of discriminating between a measurement from the data $y$ and a simulated measurement $\hat{y}$, we directly discriminate between samples $x$ from the signal distribution and the output of the reconstruction network $\hat{x}$. For a diagram describing the model, refer to appendix C.1.

### 4.3.3 PAIRED VARIANT.

This is a variant of our model where we have access to signal measurement pairs $(y, x)$ from the joint distribution $p_{Y,X}$. Given input measurement $y$, the reconstruction is obtained by regressing $y$ to the associated signal $x$ using a MSE loss. In order to avoid blurry samples, we add an adversarial term in the objective in order to constrain $G$ to produce realistic samples, as in Isola et al. (2016).

The model is trained using the same architectures as our model, and the hyperparameters have been found using cross-validation. For a diagram describing the model, refer to appendix C.2.

### 4.3.4 MEASUREMENT SPECIFIC BASELINES.

We also compare our model to baselines that where designed to remove specific corruptions.

**Deep Image Prior**   (Ulyanov et al. (2017)). Given a generator $G_\phi$ parametrized by randomly initialized weights $\phi$ and a measurement $y$, this method seeks to find a reconstruction from $G_\phi$ that is *close* to the measurement. For corruptions processes Patch-Band, Remove-Pixel and Remove-Pixel-Channel, we assume the access to the $\theta$ associated to the observations in the data (*i.e.* in this case, the mask). For more details, please refer to appendix C.

**Biharmonic Inpainting**   (Damelin & Hoang (2018)). By considering inpainting as a smooth surface extension domain, this baseline resolves a biharmonic equation to obtain a high order approximation of the image. This approximation is then extended to the missing part of the image. This method assumes access to the $\theta$ associated to the observations in the data (*i.e.* in this case, the mask).

**Total Variation Denoising**   (Chambolle (2004)). This denoising baseline aims to minimize the total variation of an image i.e the integral of the absolute gradient of the image. Reducing the total variation of the image removes unwanted detail, such as white noise artifacts while preserving important details such as edges and corners.

## 5 RESULTS

We will now present our results. First, we compare quantitatively our model with non-measurement specific baselines on CelebA. We then present qualitative results with samples from our model and these baselines. Comparisons with measurement specific baselines are presented in appendix D for the three datasets.

### 5.1 QUANTITATIVE RESULTS

We compare our model with baselines introduced in the previous section. We report mean square error (MSE) scores between the reconstructed $\hat{x}$ and the true signal $x$ used to generate the input $y$. Table 1 shows the MSE computed on the *test* set, a randomly selected subset of CelebA comprised of $40000$ images. Because the Conditional AmbientGan model is too computationally expensive, we only report the MSE on $40$ randomly chosen samples of the test set.

Table 1 – : Average mean square error of neural network based models on the test set of CelebA, for different measurement processes. The first two rows are model trained with no supervision, the last two row with additional supervision.

|  | Remove-Pixel | Remove-Pixel-Channel | Patch-Band | Convolve-Noise |
| --- | --- | --- | --- | --- |
| Conditional AmbientGan | 0.292 | 0.2829 | 0.1421 | 0.0814 |
| Our Model | **0.0414** | **0.0409** | **0.0165** | **0.0088** |
| Unpaired Variant | **0.037** | **0.0336** | 0.034 | 0.0103 |
| Paired Variant | 0.0383 | 0.0401 | **0.0147** | **0.0084** |

Quantitatively, our model performs well. Except for the Conditional Ambiant GAN, all the methods are quite similar in terms of MSE. Our unsupervised model reaches performance similar to its variants trained using additional supervision. We also note that when the aligned signal-observation pairs are not used (as in our Unpaired Variant), results are comparable – sometimes better – than when these pairs are used directly (as in our Paired Variant). This suggests that our likelihood term is sufficient to condition the reconstruction on the input signal.

## 5.2 QUALITATIVE RESULTS

However, quantitative results gives us only partial information. We now evaluate the quality of our reconstruction on three different datasets (Section 4.1). Figure 4 shows reconstructions obtained from different models on the CelebA dataset.We observe that Conditional AmbientGAN yields visually poor results, especially for the Remove-Pixel and Remove-Pixel-Channel measurement processes. We hypothesize that this is due to the large Euclidean distance between the measurements and the associated signals, and the suboptimality of the generator. Visually, the quality of our model's reconstructions are coherent with the quantitative results: they are comparable to its paired and unpaired counterparts (Section 4.3). Figures (5), (6), (7), and (8) each show reconstructions from a given measurement processe on different datasets. Our model is able to produce images with good visual quality while remaining coherent with the underlying uncorrupted images. In Figures (11), (12), (13) and (14) in appendix D, we compare our model with commonly used inpainting or denoising methods. We can see that contrary to these methods, we are able to capture semantic information from the dataset. Typically, in Figure 14, the model infers missing eyes or noses, without ever having seen them. Additional samples are available in the appendix D, refer to Figures (15), (16), (17), and (18).

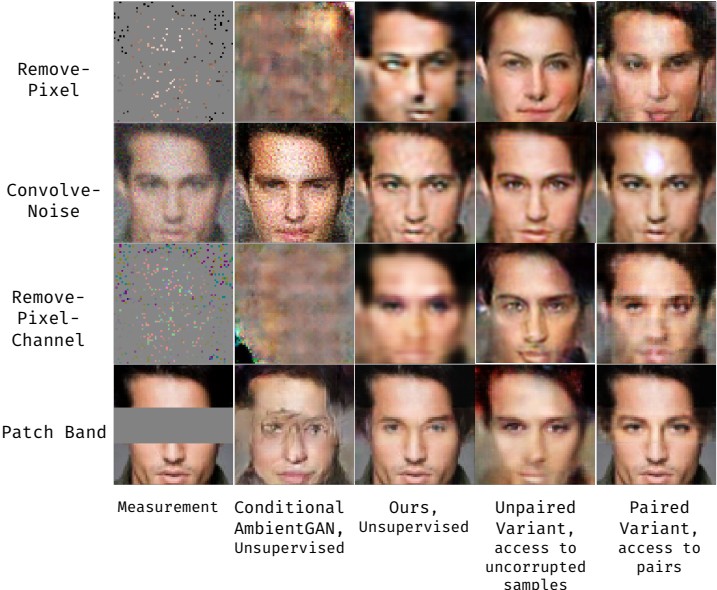

Figure 4 – Model reconstructions for different corruption processes, on CelebA. Each row corresponds to a specific corruption process, and each column to a particular model.

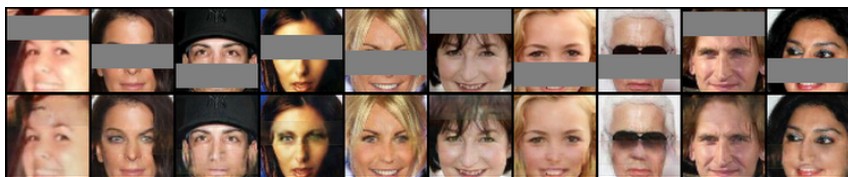

Figure 5 – On the top row, randomly sampled test set measurements from CelebA corrupted using Patch-Band($h = 20$), and below, our associated reconstructions.

## 6 RELATED WORK

To our knowledge, there is no other Deep Learning approach attempting to solve the unsupervised signal reconstruction problem. However, some of the ideas developed here are close to or even inspired from recent work.

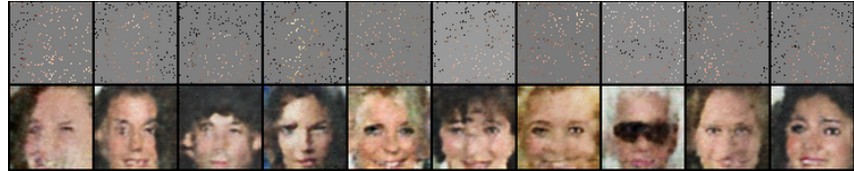

Figure 6 – On the top row, randomly sampled test set measurements from CelebA corrupted using Remove-Pixel($p = 0.95$), and below, our associated reconstructions.

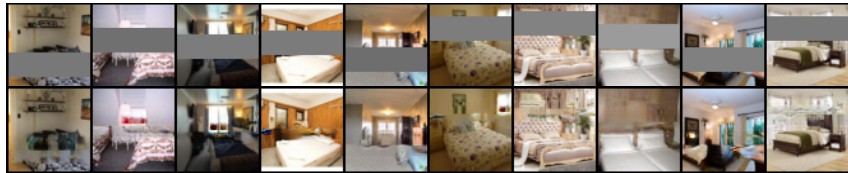

Figure 7 – On the top row, randomly sampled test set measurements from LSUN corrupted using Patch-Band($h = 20$), and below, our associated reconstructions.

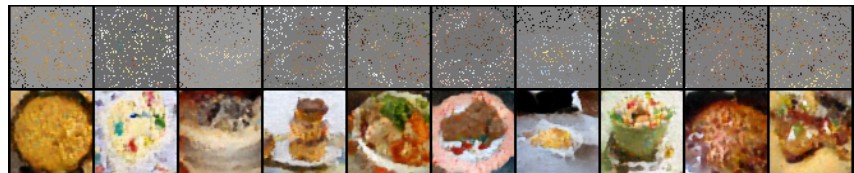

Figure 8 – On the top row, randomly sampled test set measurements from Recipe-1M corrupted using Remove-Pixel ($p = 0.9$), and below, our associated reconstructions.

In order to enforce high likelihood, we incorporate a penalty in our objective that is similar to the Cycle Consistency loss, used in several contexts (Zhu et al. (2017), Lample et al. (2017), Almahairi et al. (2018)). This constraint is used to learn from unpaired data sets. Moreover, they too use adversarial training to constrain the marginal distribution induced by the generator.

In the context of image super resolution, (Ledig et al. (2016), Sønderby et al. (2016), Mardani et al. (2017)) attempt to retrieve *maximum a posteriori* estimates of the super resolution image conditioned on an input image. They too use a generative model of the signal trained in an adversarial fashion using samples from signal distribution to constrain their reconstructions. Their approach is fully supervised.

Other works attempt to solve ill-posed inverse problems using generative models (Bora et al. (2017), Asim et al. (2018), Tripathi et al. (2018), Van Veen et al. (2018)). The general approach in all these papers consists to first train a generative model on the uncorrupted signal distribution. Then, given a measurement from which we wish to reconstruct the signal, it is *inverted* by finding the latent input code that generated the uncorrupted image, by minimizing the mean square error between the corrupted reconstruction and the measurement. This requires solving an optimization problem for each image, which takes several minutes (Ulyanov et al. (2017)) on GPU, and requires random restarts to avoid falling a bad local minima. Again, the setting is fully supervised.

Finally Lehtinen et al. (2018) propose a method for denoising images without direct supervision. They train a network to regress a corrupted image to the same image with a different corruption value. Assuming the corruption has zero-mean, their network learns to remove the corruption by the conditional expectation. This setting implicitly assumes access to the distribution of uncorrupted images in order to generate different noisy versions of the same image, which is not our case.

## 7 CONCLUSION

We have proposed a general formulation to recover a signal from lossy measurements using a neural network, without having access to uncorrupted signal data. We have formulated the problem as

finding a *maximum a posteriori* estimate of the signal given its observation, for all observations in the training set. This gives us a natural objective for our neural network, composed of a linear combination of an adversarial loss for recovering realistic signals, and a reconstruction loss to tie the reconstruction to its associated observation. Our approach yields results superior to the baselines, while staying competitive with other model variants that have access to higher forms of supervision.

For future work, we plan to apply our framework to different corruption processes, and evaluate our model's performance in real world settings, specifically for retrieving uncorrupted scientific data. Another interesting research direction would be to make our reconstruction network stochastic, in order to approximate the true posterior of the signal given the measurement, and to obtain uncertainty estimates.

ACKNOWLEDGMENTS

This work was partially funded by ANR project LOCUST - ANR-15-CE23-0027 and by CLEAR - Center for LEArning & data Retrieval - joint lab. With Thales (`www.thalesgroup.com`).

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

## A  ADDITIONAL STEPS FOR HANDLING THE LIKELIHOOD

We develop below the different steps for handling the likelihood summarized in Section 3.1:

1. Making use of the independence between $X$ and $\Theta$, we rewrite the expectation term $\mathbb{E}_{p_Y} \log p_{Y|X}(y|G(y))$ in equation (4) to $\mathbb{E}_{p_\Theta p_X p_{Y|X,\Theta}} \log p_{Y|X,\Theta}(y|G(y), \theta) + c_1$, with $c_1$ constant w.r.t. $G$:

   For all $x$, (in particular for $G(y)$), if $X$ and $\Theta$ are independent, the log-likelihood can be decomposed as:

   $$\log p_{Y|X}(y|x) = \log p_{Y,\Theta|X}(y, \theta|x) - \log p_{\Theta|X,Y}(\theta|x, y)$$
   $$\stackrel{X \perp\!\!\!\perp \Theta}{=} \log p_{Y|X,\Theta}(y|x, \theta) + \log p_\Theta(\theta) - \log p_{\Theta|Y}(\theta|y) \tag{12}$$

   Applying the expectation *w.r.t* to the joint $p_{Y,\Theta}$ on both sides, we obtain

   $$\mathbb{E}_{p_Y} \log p_{Y|X}(y|x) = \mathbb{E}_{p_{Y,\Theta}} \big\{ \log p_{Y|X,\Theta}(y|x, \theta) + \log p_\Theta(\theta) - \log p_{Y,\Theta}(\theta|y) \big\}$$
   $$= \mathbb{E}_{p_{Y,\Theta}} \big\{ \log p_{Y|X,\Theta}(y|x, \theta) \big\} + c_1 \tag{13}$$

   The terms $\log p_\Theta(\theta)$ and $\log p_{\Theta|Y}(\theta|y)$ do not depend on $x$, hence $c_1$ is a constant w.r.t. $x$. Plugging back $G(y)$ in place of $x$ and applying the law of total probabilities w.r.t. $X$ on the right hand side, we obtain:

   $$\mathbb{E}_{p_Y} \log p_{Y|X}(y|G(y)) = \mathbb{E}_{p_\Theta p_X p_{Y|X,\Theta}} \log p_{Y|X,\Theta}(y|G(y), \theta) + c_1 \tag{14}$$

2. The general measurement process in equation (1), $Y = F(X; \Theta) + \mathcal{E}$ induces $\log p_{Y|X,\Theta}(y|G(y), \theta)$ to yield a simple analytic expression:

   $$\log p(y|G(y), \theta) = -\frac{1}{2\sigma^2} \|y - F(G(y); \theta)\|_2^2 + c_2 \tag{15}$$

   with $c_2$ constant. This result is directly obtained using the fact that $\mathcal{E} \sim \mathcal{N}(0, \sigma^2 I)$.

3. The likelihood term $\mathbb{E}_{p_Y} \log p_{Y|X}(y|G(y))$ can then be replaced by $-\mathbb{E}_{p_\Theta p_X p_{Y|X,\Theta}} \frac{1}{2\sigma^2} \|y - F(G(y); \theta)\|_2^2$ in objective (4). This is because the constant $c_2$ does not change the objective.

## B  ARCHITECTURE DETAILS

**Network architecture.**  Our network architectures are inspired by the Self-Attention GAN architecture in Zhang et al. (2018). They use residual networks (He et al. (2016)), where each residual block of the generator and discriminator is comprised of 2 repeated sequences of batch normalization (Ioffe & Szegedy (2015)), ReLU activation, spectral normalization (Miyato et al. (2018)) and $3 \times 3$ convolutional layers. For the discriminator, we use the same as Zhang et al. (2018), and for reconstruction network $G$, we propose an image-to-image variant of their generator. We have not added downsampling layers: we have found that they degraded the overall model's performance. For corruption processes that yield observations that are very correlated with the input, such as Patch Band and Convolve-Noise, we have found that using $G(y) := y + \text{Net}(y)$ for the reconstruction network allows us to initialize $G$ close to identity, accelerates training and augments the overall quality of the samples.

**Hyperparameters.**  Hyperparameters have been selected on the validation set, based on the mean square error between the reconstructions $\hat{x}$ and the image $x$. As in Zhang et al. (2018), we use imbalanced learning rates for the generator and the discriminator (0.0001 and 0.0004, respectively), using the Adam optimizer (Kingma & Ba (2014)), using $\beta_1 = 0$ and $\beta_2 = 0.9$. The weights are initialized using orthogonal initialization. We set $\lambda = 2$, and exponentially decay the learning rate every 400 iterations, setting the decay factor to 0.995.

## C   ADDITIONAL INFORMATION ON THE BASELINES

### C.1   UNPAIRED VARIANT

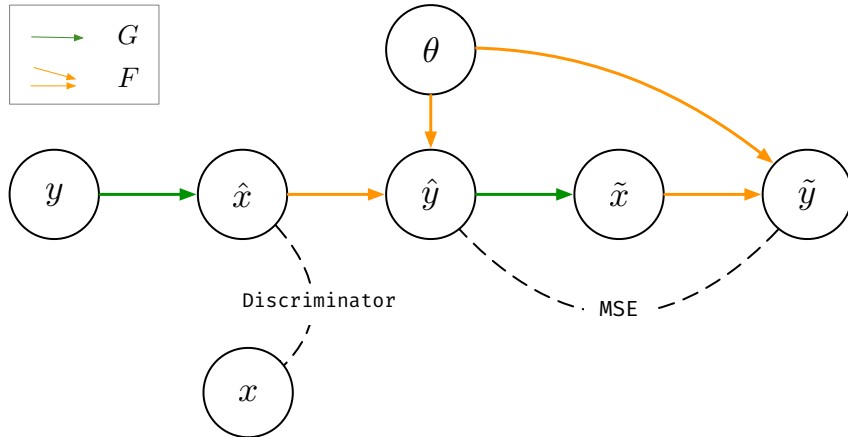

Figure 9 – Unpaired Variant of our model. As opposed to our model, this baseline has access to samples of the signal distribution $p_X$. This baseline is similar to our model, however, instead of discriminating between a measurement from the data $y$ and a simulated measurement $\hat{y}$, we directly discriminate between samples from the signal distribution and the output of the reconstruction network $\hat{x}$.

### C.2   PAIRED VARIANT

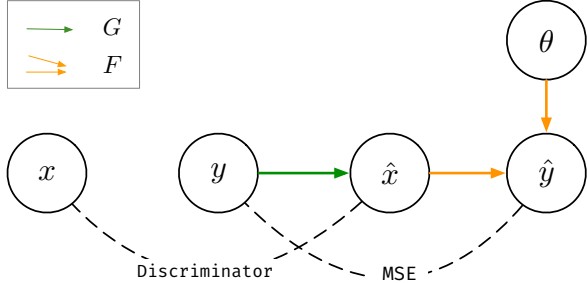

Figure 10 – Paired Variant of our model. As opposed to our model, this baseline not only has access to samples of the signal distribution $p_X$, but to signal measurement pairs $(y, x)$ from the joint distribution $p_{Y,X}$. Given input measurement $y$, the reconstruction is obtained by regressing $y$ to the associated signal $x$. In order to avoid blurry samples, we add add a adversarial term in the objective in order to enforce $G$ to produce realistic samples, as in Isola et al. (2016). The model is trained using the same architectures as the ones from our model.

### C.3  DEEP IMAGE PRIOR (DIP)

Given an input measurement $y$, a generator $G_\phi$ parameterized by random parameters $\phi$, and a random latent code $z$, the reconstruction $G_{\phi^*}(z)$ is obtained by resolving the following optimization problem:

$$\phi^* = \arg\min_\phi \|y - G_\phi(z)\|_2^2 \tag{16}$$

For measurement processes Patch-Band, Remove-Pixel and Remove-Pixel-Channel (refer to Section 4.2), the resulting reconstruction $G_{\phi^*}(z)$ was not satisfactory: $G$ was consistently regressing to the corrupted values in the measurement $y$, which led to unsatisfactory results. Instead of presenting these results, we have chosen to remove the contribution of the error terms where the measurement process induced null values from objective (16), and present the latter instead. However, this assumes access to the true value $\theta$ that corrupted datum $y$: $y = F(x; \theta)$. Formally, we resolve the following objective:

$$\arg\min_\phi \|F(y - G_\phi(z); \theta)\|_2^2 \tag{17}$$

Where $F$ acts as a mask, and eliminates the terms associated to the pixels from $y$ that have been put to 0. Note that this method corresponds to the inpainting formulation in Ulyanov et al. (2017). We used the implementation provided by the authors[4].

## D  ADDITIONAL SAMPLES

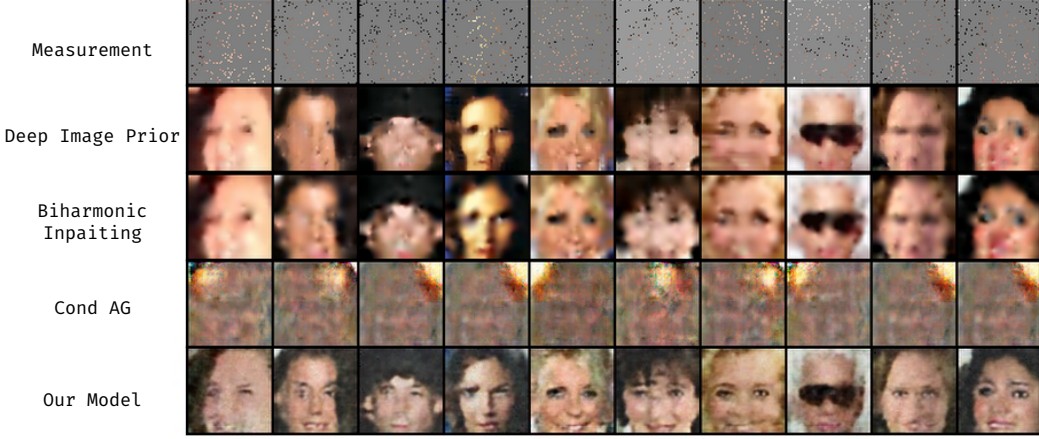

Figure 11 – Baseline comparison for the CelebA dataset. Corruption is Remove-Pixel ($p = 0.95$).

---

[4]https://dmitryulyanov.github.io/deep_image_prior

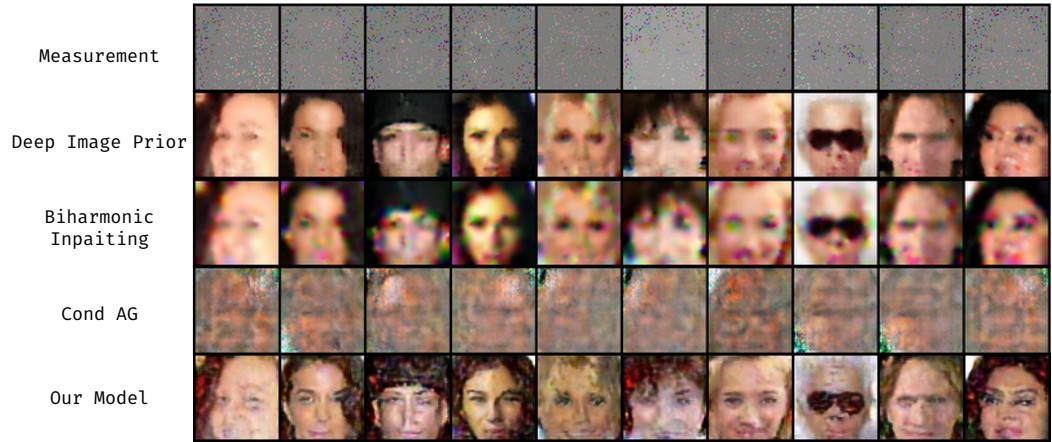

Figure 12 – Baseline comparison for the CelebA dataset. Corruption is Remove-Pixel-Channel ($p = 0.95$).

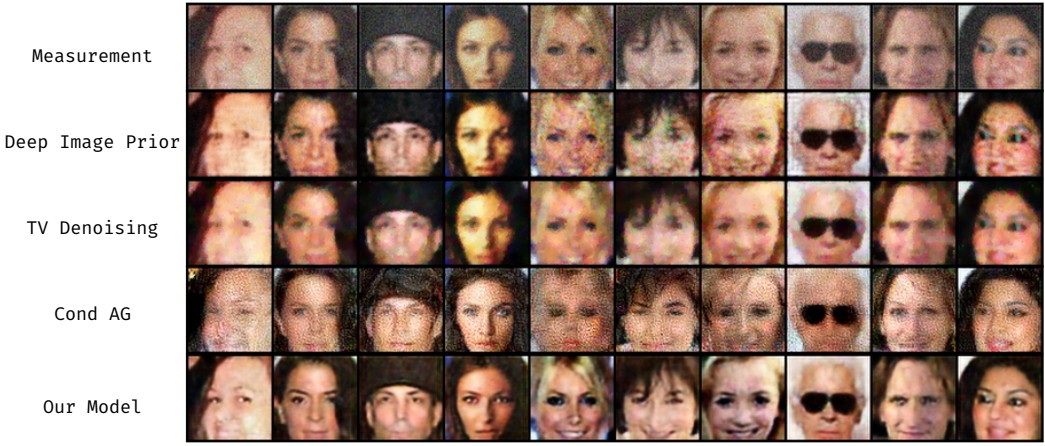

Figure 13 – Baseline comparison for the CelebA dataset. Corruption is Convnoise($\sigma_C = 0.2, l = 3$).

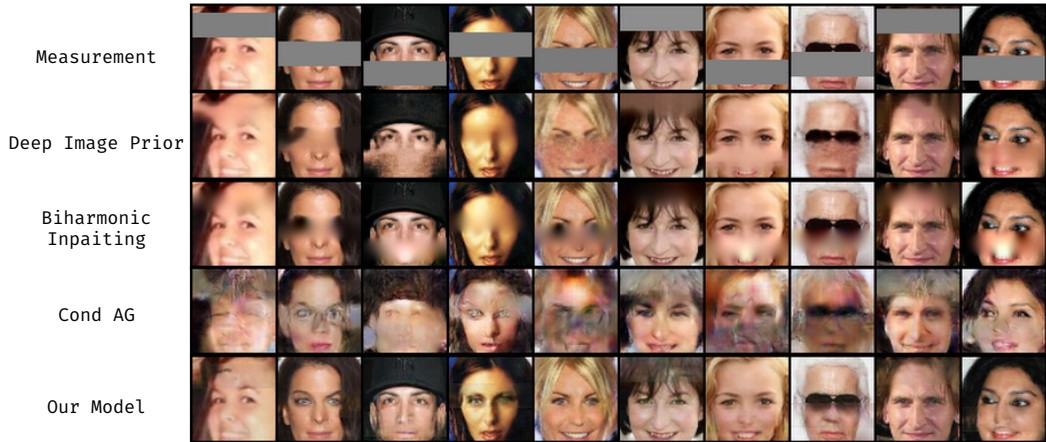

Figure 14 – Baseline comparison for the CelebA dataset. Corruption is Patch-Band($h = 20$).

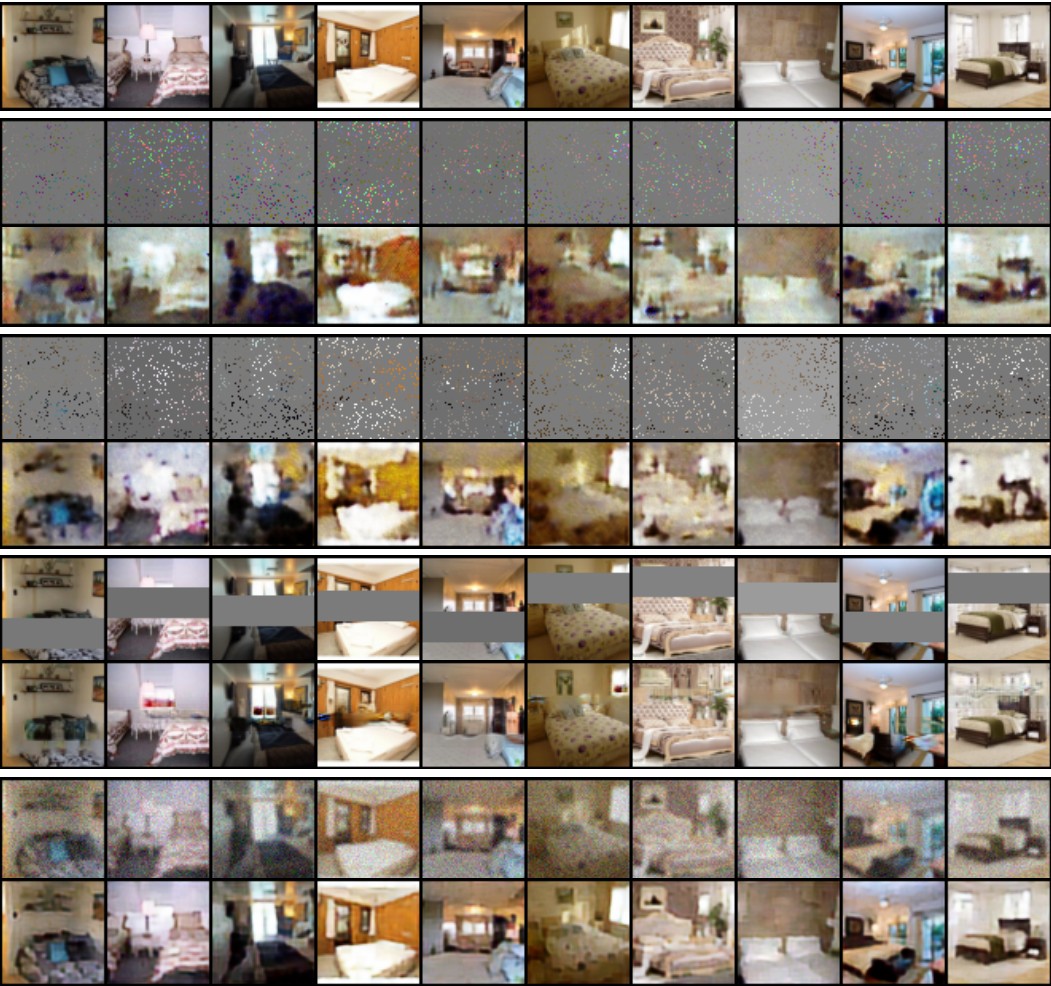

Figure 15 – On the top row, randomly sampled test set images from LSUN. Below, associated couples of corrupted observations and subsequent reconstructions from our model. From top to bottom, corruptions are Remove-Pixel-Channel($p = 0.95$), Remove-Pixel($p = 0.90$), Patch-Band($h = 20$), Convnoise($\sigma_C = 0.15, l = 3$).

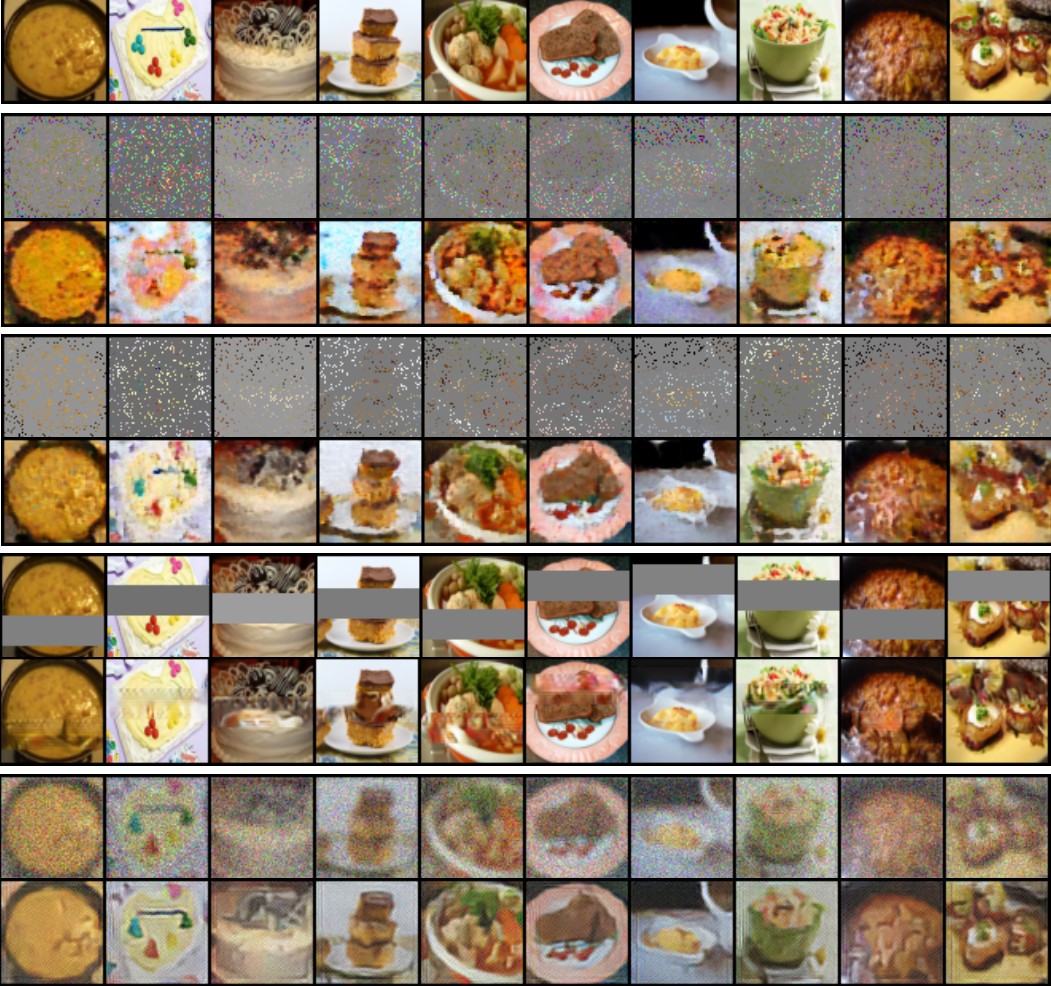

Figure 16 – On the top row, randomly sampled test set images from Recipe. Below, associated couples of corrupted observations and subsequent reconstructions from our model. From top to bottom, corruptions are Remove-Pixel-Channel($p = 0.95$), Remove-Pixel($p = 0.90$), Patch-Band($h = 20$), Convnoise($\sigma_C = 0.15, l = 3$).

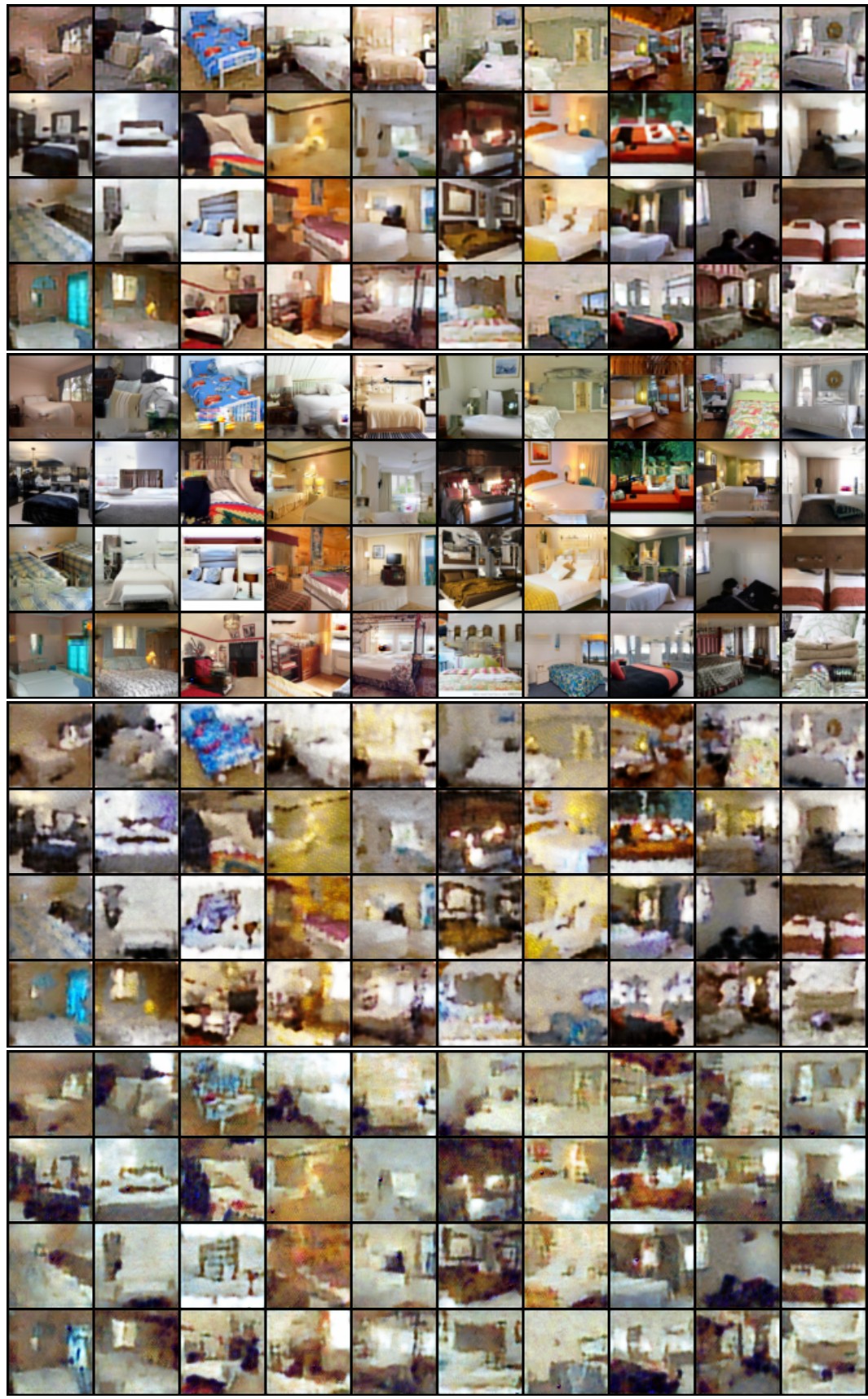

Figure 17 – Additional samples from our model of the LSUN Bedrooms dataset. From top to bottom, corruptions are Convnoise($\sigma_C = 0.15$, $l = 3$), Patch-Band($h = 20$), Remove-Pixel-Channel($p = 0.90$) and Remove-Pixel($p = 0.95$).

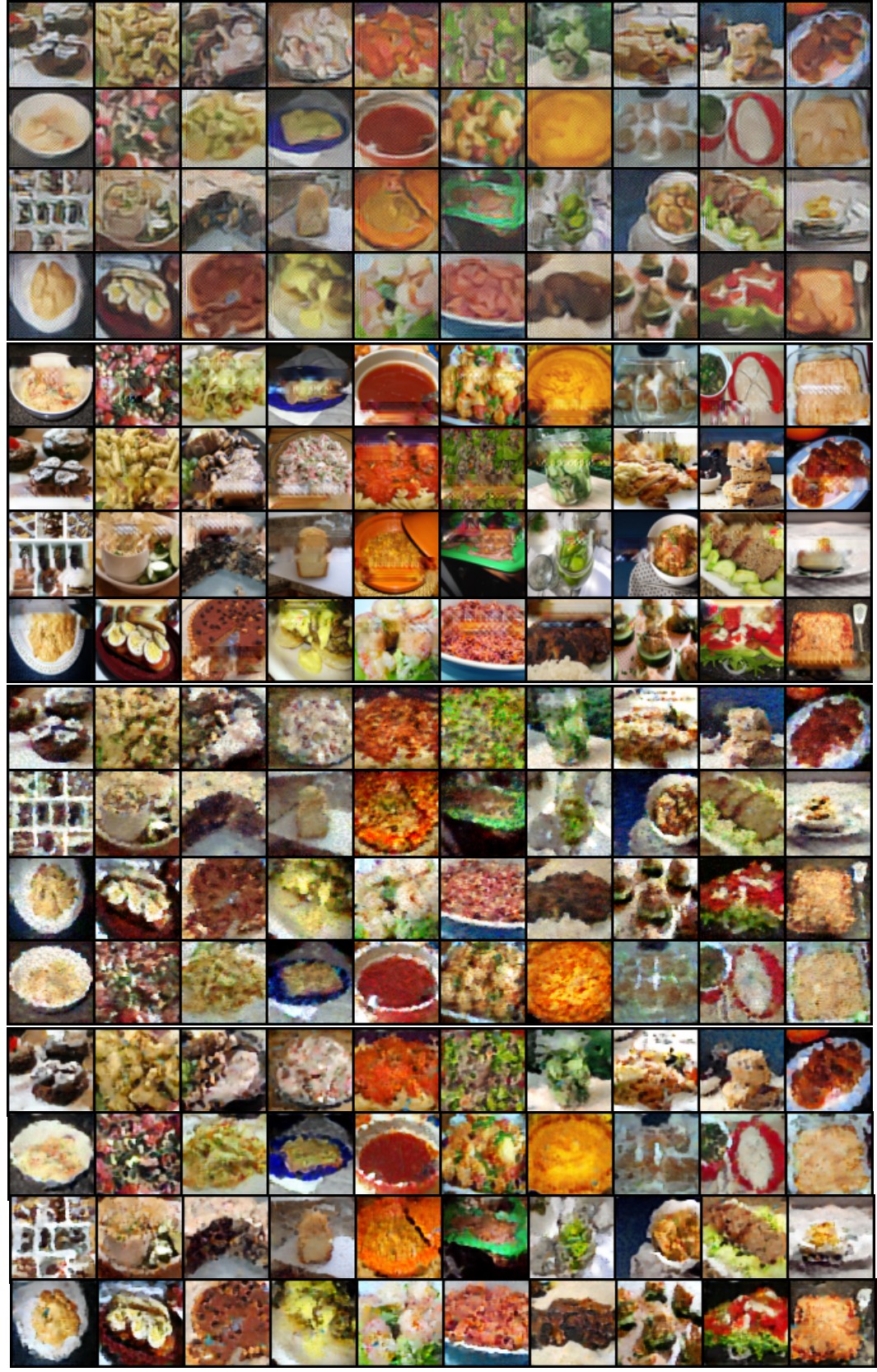

Figure 18 – Additional sample from our model, on the Recipe dataset. From top to bottom, corruptions are Convnoise($\sigma_C = 0.3$, $l = 5$), Patch-Band($h = 20$), Remove-Pixel-Channel($p = 0.90$) and Remove-Pixel($p = 0.90$).

