# OpenReview forum: "Unsupervised Adversarial Image Reconstruction"
_ICLR.cc/2019/Conference_

### Official Review · AnonReviewer2 · 2018-10-30
**Interesting but confusing**

**Rating:** 4
**Confidence:** 3

**Review:**

This paper presents a method to reconstruct images using only noisy measurements. This problem is practically interesting, since the noiseless signal may be unavailable in many applications. The approach combines ideas from recent development in compressed sensing and GANs. However, the model’s presentation is confusing, and many important details of the experiments are missing.

Pros:

* The problem is interesting and important
* The combination of compressed sensing and GANs for image reconstruction is novel

Cons:

* The model structure is unclear: for example, what is the role of the variable \theta? Section 2.1 says it is known, but the algorithm samples from its prior(?). Since there is no further explanation with respect to the experiments, I am not sure how the values of \theta or its distributions were determined. Although \theta is formally similar to the \theta parameters of the measurement function in ambientGANs, this interpretation is at odds with the example given in the paper (below eq.1, saying \theta can be positions or sizes).
* A few important details of the model are missing. For example, what is the exact structure of the measurement function F?
* The baseline models are a bit confusing. More detail about unpaired vs paired supervision would also be helpful for understanding how these baseline models use the additional information.
* Although the paper mentioned parameters are obtained from cross-validation, it would still be helpful to describe a few important ones (e.g., neural network size, weight \lambda) for comparison with other models.The experiments on only CelebA dataset are too limited.

---

> ### Author Response · Authors · 2018-11-22
> **We address the questions and add clarifications, which are also reflected in the updated draft**
>
> Thank you for the review. We are sorry that you found the overall presentation confusing, and we have been actively working on trying to make the paper much clearer. We have thus submitted a revised version of the paper taking into account your comments and answering your questions. Please see also the general comments. Typically, we have:
> *  Rewritten Section 2.1 (Problem Setting) describing the abstract measurement process and the role of theta, taking into account your comments.
> * Modified the Method section (Section 3) in order to make the explanations more straightforward and less abstract. Typically, we moved some mathematical results in the appendix for a more fluent reading.
> * Added experiments on two additional datasets: LSUN and Recipe-1M (Section 4.1 + appendix C). They illustrate the behavior of the model and of the baselines on image datasets with different characteristics and confirm the good results obtained by our model.
> * Provided additional details on the hyperparameters and the architecture for overall  reproducibility (Section 4.1). Note that we will be releasing the code shortly.
> * Added details regarding the specific measurement instances (also called corruptions) used in the experiments (Section 4.2 Corruptions),
> * Added details on the different baselines in Section 4.3. (+ Figures visually describing them in appendix )
>
> To answer your question regarding the structure of the measurement process: the measurement (or corruption) process described in equation (1) is assumed known. This means that, as in most of the problem formulations for signal recovery, the structure of the stochastic function F is known. For example, let us consider the additive Gaussian noise case. F(X, \Theta) = X + \Theta, where X is the signal random variable to be recovered, and \Theta is the noise random variable (also called corruption parameter) whose underlying distribution p_\Theta is Gaussian. This distribution  p_\Theta is assumed known, although for a specific measurement, we do not know the precise value \theta that contributed to its corruption. In other cases, typically when the measurement process induces a more structured corruption such as in our Patch Band corruption, that randomly places a band occluding the original image (introduced in Section 4.2), \Theta follows a uniform distribution taking its values from the space of pixel coordinates. To simulate this corruption process, one samples a \theta from the prior p_\Theta, and uses it to corrupt the signal x, resulting in measurement y = F(x, \theta). In this case, F places a band using \theta as the position of the top of the band. This is exactly the same formulation as the one used for AmbientGan: the associated corruptions parameter \Theta for “DropPatch” which is very similar to our “PatchBand”, corresponds to the position of the occluding patch (refer to the official implementation [1]). Note that it would also be possible to sample the size of the box, if its size varies in the corrupted data.
>
> Paired/Unpaired variant explanation :
>
>
> For the two model variants that use the additional information, *Unpaired and Paired Variant* we have added additional details in the Baseline Section 4.3, and additional Figures describing them in the Baseline appendix C. Below is an extract of the Baselines Section of the updated paper:
>
> Unpaired Variant:
> “Here, we have access to samples of the signal distribution p_X. This means that although we have no paired  samples from the joint p_X,Y, we have access to unpaired samples from p_X and p_Y. This baseline is similar to our model although, instead of discriminating between a measurement from the data y and a simulated measurement \hat{y}, we directly discriminate between samples from the signal distribution and the output of the reconstruction network \hat{x}.”
>
> Paired Variant:
> “This baseline has access to signal measurement pairs (y, x) from the joint distribution p_X,Y. Given input measurement y, the reconstruction is obtained by regressing to the associated signal x using a MSE loss. In order to avoid blurry samples, we add an adversarial term in the objective in order to constrain G to produce realistic samples, as in Pix2Pix [2]. The model is trained using the same architectures as our model, and the hyperparameters have been found using cross-validation. ”
>
>
> [1]: https://github.com/AshishBora/ambient-gan/blob/master/src/commons/measure.py#L176
> [2]: https://phillipi.github.io/pix2pix/

---

### Official Review · AnonReviewer1 · 2018-11-05
**image reconstruction from noisy samples**

**Rating:** 8
**Confidence:** 4

**Review:**

This is a very interesting paper that achieves something that seems initially impossible:
to learn to reconstruct clear images from only seeing noisy or blurry images.

The paper builds on the closely related prior work AmbientGAN which shows that it is possible to learn the *distribution* of uncorrupted samples using only corrupted samples, again a very surprising finding.
However, AmbientGAN does not try to reconstruct a single image, only to to learn the clear image distribution. The key idea that makes this is possible is knowledge of the statistics of the corruption process: the generator tries to create images that *after they have been corrupted* they look indistinguishable from real corrupted images. This surprisingly works and provably recovers the true distribution under a very wide set of corruption distributions, but tells us nothing about reconstructing an actual image from measurements.

Given access to a generative model for clear images, an image can be reconstructed from measurements by maximizing the likelihood term. This method (CS-GAN) was introduced by Bora et al. in 2017. Therefore one approach to solve the problem that this paper tackles is to first use AmbientGAN to get a generative model for clear images and then use CS-GAN using the learned GAN. If I understand correctly, this is the 'Conditional AmbientGAN' approach that is used as a baseline. This is a sensible approach given prior work. However, the authors show that their method ('Unpaired Supervision') performs significantly better compared to the Conditional AmbientGAN baseline. This is very surprising and interesting to me. Please discuss this a bit more ? As far as I understand the proposed method is a merging of AmbientGAN and CS-GAN, but much better than the naive separation. Could you give a bit more intuition on why ?

I would like to add also that the authors can use their approach to learn a better AmbientGAN. After getting their denoised images, these can be used to train a new AmbientGAN, with cleaner images as input , which should be even better no ?

In the appendix where is the proposed method in fig 5- 8 ?

Does the proposed method outperform Deep Image Prior ?

---

> ### Author Response · Authors · 2018-11-22
> **Thank your for your feedback**
>
> Thank you very much for your review and comments : they are very much appreciated.
>
> “If I understand correctly, this is the 'Conditional AmbientGAN' approach that is used as a baseline. This is a sensible approach given prior work. However, the authors show that their method ('Unpaired Supervision') performs significantly better compared to the Conditional AmbientGAN baseline. This is very surprising and interesting to me. Please discuss this a bit more ? As far as I understand the proposed method is a merging of AmbientGAN and CS-GAN, but much better than the naive separation. Could you give a bit more intuition on why ?”
>
>
> Indeed, this is correct. The conditional AmbientGan baseline combines the approaches of AmbientGan and CS-GAN. First, a generative model G of the data is learned without having access to samples of the signal distribution using the AmbientGAN framework. Then, in order to reconstruct the signal from a corrupted measurement y, we look for an input vector z of G that produces a simulated measurement G(z) that looks like y, by minimizing the Euclidean distance between G(z) and y. This method suffers from several drawbacks, which we believe can explain the poor results:
>
> * First drawback: suboptimality of the Generator. In theory, if the generator was optimal, under suitable conditions for the measurement process F, it would generate outputs belonging to the manifold of uncorrupted images (that we shall name M). Thus, projecting a measurement onto M should recover an uncorrupted image. However, this is never the case: in practice, GANs suffer from a number of problems. This means that it is possible that images from the manifold of generated images do not correspond to true samples: applying gradient descent to minimize the aforementioned distances, tend to generate  images similar to the corrupted images y, and not to uncorrupted images x. Our model does not suffer from this problem because it maximizes the log-likelihood and the prior term jointly. If G generates a signal that does not belong to M in order to maximize the log-likelihood term (similarly to what happens with the ConditionalAmbientGan baseline), the discriminator will easily be able to detect this and consequently, the reconstruction network G is corrected in order to avoid this behaviour.
>
> * Second drawback : Euclidean distance used in ConditionalAmbientGan is not adapted in the general case considered in the paper. The natural thing to do would be to find a reconstruction from M that maximizes the likelihood p(y|x). If the corruption in the measurement process corresponds to iid additive noise, it is possible to show that the problem reduces to minimizing the euclidean distance between x and y, like in ConditionalAmbientGan. However, this is not necessarily the case for other measurement processes. Indeed, in the general formulation, the likelihood is intractable;it requires marginalizing on the noise variables \theta, and for each SGD step we would need to approximate it, which would be very costly. Our likelihood term in the cost functions better reflects the true likelihood.
>
>
>
> In the appendix where is the proposed method in fig 5- 8 ?
> Fig 5-8  (now 11-14 )are samples from our baselines. The corresponding samples from our model were in figure 9 to 14. We are adding our model to figures 5-8 (11-14). Notes that we are now providing samples from other datasets (see general comments).
>
> Does the proposed method outperform Deep Image Prior ?
>
> Our experiments show that for strong corruption function DIP yields poor results compared to our model (see figure 11-14). One of the main explanation is that it does not capture semantic information from the other images of the dataset.
>
> For the measurement process Patch-Band, Remove-Pixel and Remove-Pixel-Channel, Deep Image Prior (DIP) has access to the corruption parameter \theta of the associated measurement (we have used the inpainting formulation of DIP). In other words, it has access to the mask, as opposed to our model. We have conducted experiments where DIP does not have the mask (normal formulation of DIP), and have observed very poor results (which were actually quite similar to the poor results in Conditional AmbientGAN).

---

### Official Review · AnonReviewer3 · 2018-11-05
**UNSUPERVISED ADVERSARIAL IMAGE RECONSTRUCTION**

**Rating:** 6
**Confidence:** 3

**Review:**

The authors address the problem of recovering an underlying signal from lossy and inaccurate measurements in an unsupervised fashion. They use a GAN framework to recover plausible signals from the measurements in the data.

* Authors need to test other datasets, CelebA dataset is too limited.
* Similarly, the experiment with different corruption processes are required.
* What is a definition of F. It is not clear "measurement process".

---

> ### Author Response · Authors · 2018-11-22
> **Response to comments**
>
> Thank you for your feedback. We have taken note of your comments and have been actively working to take them into account.
> You raised two main questions , one concerning the measurement process and the second one concerning the need to test the model on additional datasets.
>
> Concerning the first question, we have rewritten the sections explaining to the measurement process (please, see also the general comments about the measurement process above). Below is an extract from Section 2.1. “Problem Setting” of the updated paper version:
>
> “Suppose there exists a signal X ~ p_X we wish to acquire, but we only have access to this signal through lossy, inaccurate observation Y ~ p_Y. The measurement process is modeled through a stochastic operator F mapping signals X to their associated observations Y. We will refer to F as the measurement process, which corrupts the input signal. F is parameterized by a random variable \Theta ~ p_\Theta following an underlying distribution p_\Theta we can sample from, which represents the factors of corruption. Thus, given a specific signal x, we can simulate its measurement by first sampling \theta from p_\Theta, and then computing F(x; \theta). Additional sources of uncertainty, e.g. due to unknown factors, can be modeled using additive i.i.d. Gaussian noise \Eps ~ \mathcal{N}(0, \sigma^2 I), so that the overall acquisition process becomes:
> Different instances of F will be considered, e.g. like random occlusions, information acquisition from a sparse subset of the signal, overly smoothing out and corrupting the original distribution with additive noise, etc... In such cases, the factors of corruption \Theta might respectively represent the position of the occlusion, the coordinates of the acquired information, or simply the values of the additive noise.”
>
>
> For different measurement processes instances, also called corruptions, please refer to the Corruptions section (4.2) in the Experiments Section.
>
> As for the second remark, we have added experiments conducted on two additional datasets: LSUN Bedrooms, and Recipe1M. The results are provided in section 5 and in appendix 3. Overall this confirms the good results of the model already obtained on the first dataset.

---

### Author Response · Authors · 2018-11-22
**General Comments and Paper Revision**

Thanks to all the reviewers for their comments and suggestions. We tried to take all of them into account, we reorganized the paper accordingly and hope to provide now all the required precisions. We address below some general comments/ questions raised by the reviewers and then give detailed answers for each review.

The model presentation as been rewritten, highlighting the main ideas and results (section 3) while deferring some mathematical details to Appendix A. We have added figures illustrating the different components of the model (Fig. 1, 2, 3).
Details on the model parameters used for the experiments are provided in section 4.1, details on the corruption processes used for the experiments in section 4.2 and the baselines used for comparison are described quite extensively in section 4.3.
We performed tests on two additional datasets (LSUN Bedrooms and Recipe-1M). The three datasets have different characteristics, these experiments thus illustrate the model behavior for these different contexts. In the initial version, tests were performed on the CelebA dataset only, and two reviewers mentioned that this was too limited.
 Finally, the reviewers raised questions on the nature of the perturbation mechanism (the F(x;theta) function in the text). We agree that the description might have been unclear. This is now fully described in section 2.1. In a few words, we suppose that there exists a signal x we wish to reconstruct, but we only have access to x through lossy measurements y. The measurement process is modeled by a stochastic function with corruption parameters theta associated to a prior distribution p_Theta. The observations y are then supposed to be  generated as y = F(x; theta). We have added discussions in the text, explaining the instances of F and p_Theta associated to the different types of corruptions used in the experiments.

---

### Author Response · Authors · 2018-12-14
**Code Release**

We have released the code used in this paper : https://github.com/UNIR-Anonymous/UNIR

---

### Meta-Review · Area_Chair1 · 2018-12-16
**Good work, but a few issues should be addressed in the camera-ready version.**

**Confidence:** 3
**Recommendation:** Accept (Poster)

**Metareview:**

This paper proposes a GAN-based method to recover images from a noisy version of it. The paper builds upon existing works on AmbientGAN and CS-GAN. By combining the two approaches, the work finds a new method that performs better than existing approaches.

The paper clearly has new interesting ideas which have been executed well. Two of the reviewers have voted in favour of acceptance, with one of the reviewer providing an extensive and detailed review. The third reviewer however has some doubts which were not resolved completely after the rebuttal.

Upon reading the work myself, I am convinced that this will be interesting to the community. However, I will recommend the authors to take the comments of Reviewer 2 into account and do whatever it takes to resolve issues pointed by the reviewer.

During the review process, another related work was found to be very similar to the approach discussed in this work. This work should be cited in the paper, as a prior work that the authors were unaware of.
https://arxiv.org/abs/1812.04744
Please also discuss any new insights this work offers on top of this existing work.

Given that the above suggestions are taken into account, I recommend to accept this paper.